# The Psychoneuroimmunology of Stress Regulation in Pediatric Cancer Patients

**DOI:** 10.3390/cancers13184684

**Published:** 2021-09-18

**Authors:** Gillian E. White, Jessica E. Caterini, Victoria McCann, Kate Rendall, Paul C. Nathan, Shawn G. Rhind, Heather Jones, Greg D. Wells

**Affiliations:** 1Translational Medicine, The Hospital for Sick Children, Toronto, ON M5G 1X8, Canada; gillianwhite018@gmail.com (G.E.W.); jcaterini@qmed.ca (J.E.C.); kate.rendall@mail.utoronto.ca (K.R.); 2School of Medicine, Queen’s University, Kingston, ON K7L 3N6, Canada; vmccann@qmed.ca; 3Division of Hematology/Oncology, The Hospital for Sick Children, Toronto, ON M5G 1X8, Canada; paul.nathan@sickkids.ca (P.C.N.); heather.jones@sickkids.ca (H.J.); 4Defence Research and Development Canada, Toronto Research Centre, Toronto, ON M3K 2C9, Canada; Shawn.Rhind@drdc-rddc.gc.ca; 5Faculty of Kinesiology & Physical Education, University of Toronto, Toronto, ON M5S 2W6, Canada

**Keywords:** stress, allostasis, stress regulatory pathways, stress dysregulation, stress reactivity, early life stress, childhood cancer, cytotoxic cancer treatment, interventions

## Abstract

**Simple Summary:**

There are many commonalities between children with cancer and other populations that experience early-life stress. Thus, it is important to review the existing research surrounding the stress response in the pediatric cancer population. In this review, we describe the psychoneuroimmunology behind stress regulation and the differences observed in stress regulatory pathways in childhood cancer patients. Our objective is to provide a clinically relevant summary of the stress pathways contributing to, and exacerbating, childhood illness and outline some potential interventions.

**Abstract:**

Stress is a ubiquitous experience that can be adaptive or maladaptive. Physiological stress regulation, or allostasis, can be disrupted at any point along the regulatory pathway resulting in adverse effects for the individual. Children with cancer exhibit significant changes to these pathways in line with stress dysregulation and long-term effects similar to those observed in other early-life stress populations, which are thought to be, in part, a result of cytotoxic cancer treatments. Children with cancer may have disruption to several steps in the stress-regulatory pathway including cognitive-affective function, neurological disruption to stress regulatory brain regions, altered adrenal and endocrine function, and disrupted tissue integrity, as well as lower engagement in positive coping behaviours such as physical activity and pro-social habits. To date, there has been minimal study of stress reactivity patterns in childhood illness populations. Nor has the role of stress regulation in long-term health and function been elucidated. We conclude that consideration of stress regulation in childhood cancer may be crucial in understanding and treating the disease.

## 1. Introduction

Stress is a ubiquitous experience with significant impacts on health and function. Stress refers to the physiological state of the body in response to a stressor, whereas the stressors themselves are challenges, threats, demands and constraints that provide barriers to the normal daily functioning of the individual, thereby inducing stress and its related consequences on the body [1]. Stress responses function acutely to defend homeostasis during changing internal and external demands [2,3] and transduce external stimuli into physiological signals to support effective navigation of the environment and encode relevant information for future events [4]. Depending on the nature of the stressor and the interpretation of it, stress can be adaptive (“eustress”), broadening one’s ability to cope with environmental conditions and challenging experiences effectively, or maladaptive (“distress”), straining one’s ability to cope and contributing to general dysfunction [2,5,6].

Significant or chronic exposure to stress activation can cause long-term changes to how the stress-regulatory system responds to future stressors [6,7,8]. This is particularly true of significant stress exposure during times of developmental plasticity, such as childhood and adolescence [9,10,11,12,13]. Several chronic diseases in adults and children, including cardiovascular disease, metabolic conditions, cancer and immunologic conditions have been attributed to chronic stress and/or dysfunctional stress regulation, in addition to a number of psychological and cognitive conditions and nonclinical functional outcomes following significant or chronic stress exposure [5,14,15,16]. Risk and prevalence of these stress-related adverse outcomes have been reported in populations of children experiencing significant stress or trauma [17] and are thought to be a result of alterations in stress regulatory pathways which can be detected by examining patterns of physiological signalling in response to an acute stress exposure [5,14,15,16].

While the adverse outcomes of childhood trauma and chronic stress have been well documented, current definitions of Adverse Childhood Events (ACEs) and early life stress models from which this research draws [4,9,10,13,15,17,18] do not include childhood illness. Stress dysregulation may provide a valuable paradigm through which to understand long-term health and wellbeing in populations experiencing childhood illness, given psychological strain and physically aggressive medical treatments.

Approximately 1 in 300 children will develop cancer between birth and 20 years of age [19] which involves significant psychological burden, physical distress and barriers to engaging in positive stress coping behaviour such as pro-social and physical activity habits. Children with cancer and survivors show evidence of neurological [20,21,22,23,24,25,26,27], physiological [28,29,30,31,32,33,34] and psychosocial [24,25,27,35,36] changes potentially attributable to stress dysregulation (see Figure 1). Further, they report higher risk of late effects similar to those reported in other early life stress populations, suggesting that stress dysregulation may be worth investigating as a putative pathway, or as a modulator of the damage caused by cancer treatment. Psychoneuroimmunology, a field of study that is inherently implicated in stress regulation, examines the relationships between human behaviour, the nervous and endocrine systems, and the immune system [37]. This review describes the psychoneuroimmunology of stress regulation and support for potential clinical relevance in childhood cancer patients and survivors [38,39,40].

## 2. Stress Regulation & Allostasis

Selye defines stress as the body’s adaptive response to a “noxious agent”, involving a deviation from resting state, or homeostasis [2]. Homeostasis refers to the maintenance physiological variables essential for human life, such as internal body temperature, or pH, within a precise range [3]. Physiological stress regulation, also termed ‘allostasis’, is an adaptive process aimed at keeping the body’s systems in physiological ranges, or maintaining homeostasis, despite changing internal and external environmental conditions [3]. Stressors can be both biological or psychological, and thus can be real or perceived, past, current, anticipated or recalled [41]. Since biological stressors are typically internal, they evoke a direct physiological response without the engagement of higher order cognitive processing [42]. Psychological stimuli, however, must undergo interpretation, and thus elicits a physiological stress response indirectly [14,43,44,45]. Therefore, a psychological response is mounted in response to the perception of stress, rather than the stressor itself. For the purposes of this review, we will only consider psychological stressors and the allostatic processes that they evoke.

The allostatic processes are an integration of subjective, often subconscious appraisal of a physical or psychological stimulus, which is converted into neural and endocrine signalling in turn activating target organs that elicit an allostatic response as shown in Figure 2. Effective allostatic responses are characterised by rapid upregulation of signaling hormones and effector tissue activation, commensurate with the level of threat encountered, and rapid termination upon cessation of stimulus—producing maximal physiological response to effectively defend homeostasis during stress experience with minimal exposure to signalling biomediators and associated energy costs [46].

Allostasis is considered to be a mechanism by which the body can maintain homeostasis in the face of variable environments—stability through change [47]. To this end it can be considered from an energy conservation perspective in which the physiological responses are predictive and aim to maximise energy efficiency in coping with environmental demands [48]. It can also be considered a calibrating mechanism in which the mediators of the response feedback to the earlier steps to adapt optimally to environmental contexts [4]. The hypothesis of allostasis as a mechanism in long-term health and functional outcomes in childhood cancer patients and survivors is rooted in the data indicating altered regulation of the cognitive, psychological, physiological pathways that regulate physiological responses to stress. We suggest that understanding and mitigating this dysregulation of physiological responses to stress is integral to improving care in this population. Therefore, for the purposes of this paper, we take the view of allostasis as the process by which the body responds physiologically to stressors in order to regain homeostasis.

Acutely, allostasis is adaptive and functions to maintain homeostasis in face of a prospective disruption by mediating a fight or flight response [8]. Biomediators of allostasis, specifically cortisol, also provide feedback to brain regions responsible for triggering allostatic responses in a self-regulating process that tunes the response to the stressor presented and terminates the acute activation of allostasis [6,7,8]. This same biochemical feedback calibrates the system to future stress exposures through both neurological encoding and affective working memory [4,6,8].

There are important functional consequences of chronic allostatic stress. The catabolic and energy costly processes needed to mount allostatic responses contribute to wear and tear on target organs, potentially exacerbating underlying weakness or dysfunction in these tissues. Further, the chronic feedback of biomediators on stress regulatory corticolimbic brain centres, particularly the hippocampus, alter their structure and function, ultimately impacting the overall regulation and responsivity of the system [49,50,51,52,53,54]. Due to the self-regulatory nature of stress regulation, dysregulation can occur with disruption to any point in the pathway shown in Figure 2. Typically, dysregulation that contributes to stress-related health and dysfunction is characterised by one of four patterns of stress reactivity: (1). Overly frequent activation; (2). No habituation to familiar stimuli; (3). Inefficient termination; or (4). Hypoactivation of one pathway resulting in compensatory overactivation of the other [3].

While stress can contribute to the development or acceleration of illness, it does not cause illness or dysfunction per se. Rather, the adaptive function of allostasis requires systemic upregulation. Often these are catabolic processes, resulting in wear and tear over time or “allostatic load” [3,10,18,55,56], which describes the cost that this prolonged upregulation and activation of compensatory effectors has on the body [57]. Allostatic load exacerbates existing tissue weaknesses, whether hereditary genetic predispositions or susceptibility from a previous insult, or a concomitant one such as cancer therapy. The aggregate of this subclinical dysfunction can have significant health and functional implications and has been associated with all-cause morbidity and mortality [3,55].

Much research to date has focused on individuals who experienced significant childhood adversity or ACE’s (such as physical or emotional child abuse, neglect, parental substance abuse, household and family turmoil, etc.) but leaves out childhood illness [4,7,10,11,17,18,58]. Many commonalities exist between populations experiencing psychosocial early life stress and children experiencing childhood illness [13,17,18,28,58,59,60]. This presents the possibility that children treated for illness many be susceptible to long-term health and dysfunction associated with altered stress regulation. Childhood cancer patients experience significant stress prolonged over the duration of treatment, as well as significant physical and psychosocial effects of treatment and disruptions to their normal developmental opportunities [23,61]. As a disruption to any step in the pathway can result in dysfunctional regulation, we will review the role of each step and evidence of prospective dysfunction in childhood cancer patients and survivors.

## 3. Cognitive-Affective Appraisal

Allostatic processes are regulated by the integration of inputs to corticolimbic brain regions including the prefrontal cortex (PFC), hippocampus, amygdala and brainstem. These regions are implicated in many other functions besides stress circuitry such as decision making and higher cognition, learning and memory, emotional processing and the judgement of salience, among others [62]. Neurocognitive and neuropsychological function development corresponds with the timing of brain development through childhood and adolescence. In brief, functions associated with emotional reactivity develop ahead of those necessary for cognitive reappraisal and self-regulation during childhood [12,63], in such a way that the time-lag in development in conjunction with social contexts of adolescence is often attributed to the behavioural trends (impulsivity, risk taking) and greater stress reactivity seen during adolescence [12,64].

For evolutionary reasons, stimuli that involve social judgement, are goal oriented and include unpredictability, novelty and uncertainty are most salient and reliably evoke a physiological response [65,66]. The subjective nature of cognitive-affective appraisal presents significant inter-individual variability and is influenced by lived experience, disposition, cognitive and psychological strategies, as well as positive coping behaviours such as physical activity and social connections. Human and animal models of chronic stress have shown impairments to memory and executive functions, and these also appear to be long-term deficits reported in children and adults who have experienced early life stress [20,49].

Studies investigating the neurocognitive consequences of chemotherapy during childhood (see Figure 1) have also found impairments to working memory and executive function [20,67,68,69], as well as processing speed, task efficiency, attention, memory and learning [70,71,72]. An important association has been made between these functions and self-regulatory behaviour, such as effective coping [73]. As noted by Campbell et al. (2007), the consequences of these stress-induced impairments are present not only in school settings, but in many other domains of life such as social relationships, emotional control, coping skills, the workplace and overall quality of life [70]. In line with this, Krull et al. (2013) found that over 10 years post-treatment, 28–59% of childhood cancer survivors reported neurocognitive and neuropsychological impairments, the greatest being reduced attention and executive functions, which were most closely associated with treatment with dexamethasone, a synthetic glucocorticoid, when excluding those who received cranial radiation therapy [35]. They further found that survivors reported an approximately 5% annual increase in self-reported behavioural issues related to self-regulation, which impacted functioning in academic and occupational settings [35].

Additional risk to effective cognitive-affective stress appraisal may be conferred by lower engagement in stress buffering behaviours such as socialisation and physical activity that promote adaptive coping [74,75,76]. Indeed, in a recent study of childhood Acute Lymphoblastic Leukemia (ALL) survivors, correlations were found between low levels of physical activity and inattention [77]. Further, disruption to corticolimbic brain regions central to these interpretive processes has been reported in several childhood cancer populations and is thought to be a result of cytotoxic treatments including chemotherapies that are neurotoxic and/or cross the blood brain barrier.

## 4. Neurological Triggering

During threat appraisal, corticotrophin-releasing hormone (CRH) is released from the hypothalamus and activates the hypothalamic-pituitary-adrenal (HPA) axis, while norepinephrine from the Locus-Coeruleus activates the autonomic nervous system (ANS), and ANS input directly activates target organs, and the sympatho-adreno-medullary (SAM) pathway [78]. The glucocorticoid cascade hypothesis posits that significant or enduring stress exposure will result in excessive cortisol exposure leading to altered hippocampal functional control over HPA scaling and termination, which in turn leads to further cortisol exposure and propagation of dysregulated stress signalling, ultimately leading to adverse health and functional effects [79]. Critically, the biomediators released by neurological triggering (cortisol and catecholamines) feed back to the brain, influencing both the cognitive-affective experience of stress, as well as continued neurological triggering. Cortisol feedback in particular is thought to be critical to tuning and terminating the activation of the HPA axis, by influencing neurotransmission of the hippocampus. Cortisol-mediated neuroplastic and neurotransmission changes to the hippocampus are commonly thought to be integral to the development of stress dysregulation. This may be of particular interest to populations treated with pharmacological glucocorticoids, such as prednisone and dexamethasone, both of which have CNS penetrance and are used for CNS prophylaxis for pediatric leukemia patients [23].

The protracted nature of brain development is such that subcortical and limbic structures (amygdala, hippocampus, brain stem) development precedes prefrontal and frontal cortex development [63,80,81]. Even though most adult hippocampal networks are apparent during childhood, their connections to lateral lobes increase throughout childhood (ages 4–10) [81] and PFC development is not fully developed until early adulthood [82].

Significant evidence in both human and animal models have found changes to neurological structures responsible for cognitive-affective processing and neurological triggering to be affected by exposure to chronic stress or early life stress [18,49,83,84,85,86,87,88]. Chronic stress in animal models has shown reduced hippocampal volume [83], reduced neurogenesis in the hippocampus [84,85,86], PFC atrophy [49] and amygdalar hypertrophy [49], concomitant with altered cognitive functions associated with those regions [49,83,84,85,86]. Human studies have found similar changes in altered hippocampal structure and function [18,87], PFC impairment [18,88] and other neurological impairments [88] in populations experiencing early life stress. Some inconsistencies exist in neurological studies of children, which have been attributed to the protracted nature of neurological development, such that the full extent of impact of early life stress on regional brain structure and function does not become apparent until the third decade of life [12,81,89]. It follows that the nature of brain changes is also sensitive developmentally, such that the regions that are in development when the stress occurs are most likely to experience long-term dysfunction [61,90].

Disruption to corticolimbic brain regions critical to stress regulation have been reported in childhood cancer patients treated with chemotherapy and/or cranial radiation. It is important to note that most studies examining the relationship between chemotherapy and brain structure and function changes have been performed mostly using cross-sectional study designs, as detailed in Table 1. Many of the neurological consequences associated with chemotherapy, including lower white matter volume [22,91,92,93], altered hippocampal microstructure [84,86,94] and altered PFC microstructure [73], are likely to impact neurological triggering and feedback effects on cognitive-affective appraisal. Both human [67,94] and animal [84,85,86] studies have reported lower hippocampal volumes and impaired neurogenesis related to various chemotherapeutic agents, similar to findings in other early life stress studies [18] (see Table 1). Amygdala changes have also been reported in adult cancer populations [26,94,95] and have been related to adverse psychological effects [26,95], and recently, reduced amygdala and dorsal striatum brain matter volume has been found in pediatric cancer populations [96]. Chemotherapy-treated survivors of childhood cancer displayed lower cerebellar volumes, versus healthy controls, which was associated with both poorer performance on neurocognitive testing and exposure to dexamethasone [97]. Despite some evidence of altered limbic and subcortical structure and function following chemotherapy, the majority of studies report differences in frontal and pre-frontal brain structures, as well as impairments in the function of these structures [20,21,35,67,70,98,99,100]. This may be due to the importance of these regions in global intellect and other neurocognitive deficits that have taken priority in this research space. The functional implications of these brain changes have not been studied in the context of stress, however, adrenal insufficiency and HPA dysfunction has been reported in children receiving cranial radiation suggesting that disruption to corticolimbic brain regions can have downstream effects on physiological stress signalling.

Whether these neurological changes are a direct cause of anti-neoplastic treatment (i.e., glucocorticoids, intrathecal methotrexate, cranial radiation) or endogenous stress processes, all can disrupt effective stress regulation and therefore, may be relevant to supporting long-term health and function in childhood cancer patients and survivors.

## 5. Physiological Stress Response

The signalling response evoked by neurological triggering is comprised of three distinct but overlapping systems. Brainstem activation of the ANS occurs almost immediately via neural inputs to visceral target organs activated for allostasis as well as through adrenal stimulation of catecholamines into circulation (SAM), while the hypothalamus activates the HPA axis via stimulation of the pituitary to release ACTH. These two arms of allostatic control, SAM signalling via catecholamines and HPA signalling via glucocorticoids, are responsible for the physiological and affective experience of stress.

Secondary signalling by inflammatory factors is also implicated in short and long-term effects and experiences of stress. The biomediators of inflammation, cytokines, can be produced by neutrophil demargination and activation of immune cells [109]. In the short term, this primes the immune system to protect the body from impending injury but in the long-term can contribute to reduced immune function, worse tissue healing and chronic inflammation and associated physiological and psychological disturbance [79,110].

Basal HPA and HPA-reactivity increases with age, with a marked increase around puberty (—13 for girls, 15 for boys) [64,111,112]. This is likely due to the effects of changing environmental demands, developing neurological structures that enable relevant neurocognitive and neuropsychological functions (i.e., goal-oriented behaviour is governed by the PFC and is thought to be an essential component of psychological stress provocation), and hormonal changes associated with puberty [88,111,113].Stress response patterns can be used to infer how the whole system is functioning and whether an individual appears to have a resilient or vulnerable stress phenotype. These acute patterns are thought to be demonstrative of longitudinal stress regulation and prospective allostatic load and associated consequences for health and wellbeing. In a resilient and optimised system, the response pattern should show an immediate increase with stressor onset followed by a rapid termination upon resolution or cessation of stress exposure. Unnecessary activation of stress responses overexposes corticolimbic brain regions to the neuroplastic and neurotransmission effects of biomediators. Children who have experienced significant adversity show divergence in their stress reactivity patterns [3]. Some have higher basal activation and hypoactivity to acute stress exposure in one or multiple signalling systems, while others show hyperactivity and impaired termination. Elevated inflammation is also a common finding that is a purported mechanism of long-term adverse health and functional outcomes in chronic stress populations [32], the degree of which is different to that of an inflammatory response to infection. Acute stress reactivity to psychological stimuli has been used in a variety of pediatric populations to predict stress-related risks for health and wellbeing [64,111,114,115,116,117,118] and it is generally considered that any deviation (hypoactivation or hyperactivation), is likely indicative of dysfunctional signalling and regulation [16,58].

Few studies have examined stress reactivity patterns in childhood illness populations or considered the role of stress regulation in long-term health and function, despite the significant psychological distress in addition to the direct physiological effects of medical treatments. This is even more surprising considering that many childhood illnesses require the use of synthetic glucocorticoids for treatment. Thus, little evidence for the chronic effects of stress exposure exists in childhood cancer, although some studies have reported that elevated inflammation and oxidative stress persists post treatment [31,102]. Kennedy (2005) found total antioxidant capacity to be lower in ALL patients 6 months post-treatment, and this was correlated with better clinical outcomes including lower rate of infection and hospitalisation, higher quality of life and better treatment tolerance [31]. Similarly, Mazur et al. (2004) found elevated circulating cytokines, TNFα, IL-2 and IL-8, in ALL patients 3, 6 and 12-months post-treatment [32]. Importantly, studies of adult patients have shown a relationship between cytokine status and neurocognitive function [35] providing support for a link between physiological activation and cognitive-affective capacities. However, no research has explicitly investigated physiological stress response profiles, which might provide clues into mechanisms of adverse long-term and late effects in this population. As altered acute stress profiles and biochemical signalling is considered the mechanism linking stressful experiences to long-term health and dysfunction in other populations experiencing significant stress, this is an important area for further research in childhood cancer patient and survivor populations that may provide insight into late effects of childhood cancer [119].

## 6. Target Organ Activation

The culmination of stress responses is in allostatic processes preparing the body for a threat and defending homeostasis in the face of changing environmental conditions. However, what was an adaptive response for physical stressors evolutionarily may not be effective for psychological stressors of contemporary lives. The inappropriate activation of stress systems is thought to contribute to a wide range of illnesses reflecting allostatic load [55].

Allostatic load, the aggregation of these subclinical issues or progression of a subclinical issue into a clinical issue over time, is associated with significant risk of morbidity and mortality [3,120]. Children who experience significant life stress have higher risk of heart disease, diabetes, cancer, chronic lung disease, skeletal fractures, liver disease, mental distress disability and overall worse health ratings [17,121]. At the tissue level, evidence of oxidative stress including cellular aging [122,123,124] and shortened telomeres [125,126] indicate systemic tissue disruption by early life stress.

The consequences of early life stress and chronic stress is further demonstrated by the higher rates of morbidity, as cellular vulnerabilities result in dysfunction, ultimately increasing risk of all-cause mortality in these populations [5]. These same issues are common side-effects observed in children’s cancer treatment, especially those treated with synthetic glucocorticoids [28,60,103].

In line with other populations of children who have experienced significant stress, childhood cancer survivors have also been reported to have shortened telomeres, and this has been associated with higher chronic inflammation [104] and higher incidence of late effects during survivorship [105]. Pediatric cancer survivors differ from age-related controls in terms of activation of the adaptive immune system, chronic, low-grade inflammation, as well as immune tolerance resulting from the synthesis of immunomodulators via the tryptophan-kynurenine metabolic pathway [127]. These changes resemble an aging phenotype observed in older populations [128] and are indicative of allostatic load [127]. Some research shows that pediatric cancer survivors have increased biological age relative to their chronological age, as indicated by shortened telomeres [104,105], epigenetic age acceleration [106] and biochemical and molecular markers such as inefficient oxidative phosphorylation, increased lipid peroxidation and decreased expression of metabolic proteins and those involved in mitochondrial biogenesis [107]. Childhood cancer survivors also report an increased incidence of premature frailty associated with radiation treatment [108]. The cytotoxic nature of childhood cancer treatments can cause significant damage and disruption to developing organ systems, which may present more inherent vulnerability to stress exacerbation than the general population. Childhood cancer patients have a higher risk of many chronic illnesses including cardiovascular disease, secondary cancer, metabolic conditions, depression and anxiety, as well as subclinical but lower reported health related quality of life [24,28,60,101]. Dysrhythmias and other indicators of cardiac dysfunction and conductive symptoms are reported in adult survivors of childhood cancers [59], which is most commonly attributed to anthracycline exposure used for anti-neoplastic treatment [129], or radiation to a field that involves the heart. In a population of children and youth who survived mixed types of cancer, 28.2% were reported to exhibit hypertensive or pre-hypertensive signs [130]. Similarly, Cardous-Ubbink (2010) found increased risk of hypertension in adult survivors of childhood cancer, related to BMI, cyclophosphamide, cisplatin or abdominal radiation [33]. In a population of adult survivors of childhood cancer, stress and distress were associated with adverse cardiovascular health conditions such as hypertension, dysrhythmia, dyslipidemia and metabolic syndrome [131]. Even though stress reactivity and function of the systems involved in stress responses have not been investigated in childhood cancer patients or survivors, many of the conditions experienced by this population over their lifetime are those that can be developed or worsened by dysfunctional stress regulation [3,28,60].

It is not possible to differentiate between the contribution of stress dysregulation of target organs over and above that caused by direct cytotoxicity of anti-neoplastic treatment. However, it is still relevant to reducing the burden of illness, as stress dysregulation effects on tissue integrity and function are self-propagating and many adverse health and functional outcomes worsen with time since treatment in childhood cancer survivors. Thus, it is possible that these work in concert to contribute to adverse late-effects, with direct effects of treatment producing vulnerabilities and initial weakness, and stress and other long-term pathways contributing to worsening of function with time. For example, direct disruption to corticolimbic brain regions mediating upstream stress regulation can be further exacerbated by altered stress signalling and may be important targets for intervention to promote better health and function during survivorship.

## 7. Considerations for Interventions

Each step in the stress regulatory pathway can alter the function of the system acutely and over time influence the potential of stress to contribute to adverse health and dysfunction. Potential intervention options to reduce the burden of childhood illness centre around stress-buffering behaviours and may include encouraging social connection [132,133], pro-social behaviour [133] and physical activity [75,134,135], while teaching effective coping strategies. Mutable individual factors related to disposition and behaviour can also have a significant impact on acute physiological activation to a stress stimulus and the effects of stress on long-term health and function. The mutable factors, such as intrapsychic coping strategies, pro-social behaviour and physical activity habits, should be considered clinically meaningful as it relates to any contribution of stress dysregulation on long-term health and function [136]. Programs promoting social connections and development can be expected to have psychosocial and psychobiological benefits through positive changes to cognitive-affective processing and neurological triggering. Expectations of negative social judgement reliably provoke physiological stress responses [65,66]. Strong social connections may reduce negative expectations during cognitive-affective appraisal, reducing physiological activation of stress systems [132]. Further, social support has a strong influence over acute stress reactivity due to the release of oxytocin in the brain, inhibiting CRH production, thereby reducing neurological triggering of the HPA axis and subsequent physiological and affective experiences of a given stressor. Outcomes of social programs in clinical populations rarely focus on clinical indicators, however, positive effects of social support are reported improve stress management [132,133,137].

Several studies have shown that parents of children with cancer have a higher incidence of post-traumatic stress disorder and related symptoms when compared to parents of healthy children [138,139,140], and that these symptoms were associated between the parents and their children [139]. This suggests that the parents’ stress of having a child with cancer may have consequences for the children themselves. Parental stress has been found to be a significant predictor of functional impairment in childhood cancer survivors [141], and childhood cancer survivors may experience different parenting styles including parental overprotection due to stress [142,143]. However, these findings are not conclusive as some research has suggested parenting styles are not different from children without a history of serious illness [144]. These findings indicate the need for further investigation and potentially psychological interventions in childhood cancer patients and parents alike.

Programs promoting physical activity can be expected to have neurobiological and psychobiological benefits through positive changes to cognitive-affective processing, neurological triggering, as well as stress signalling and impacts on target organs [134,135]. Physical activity and fitness both have adaptive effects on stress reactivity and can influence the stress regulatory pathways at multiple steps. Acutely, activity promotes positive mood, reduces negative affect and alters dopamine-GR signalling [75]. Chronically, physical activity promotes executive function, increases neurogenesis of the hippocampus and reduces inflammation as well as promotes healthy function of many of the organ systems of allostatic responses. Together, the effects of physical activity can be expected to reduce acute activation, promote habituation to future stressors and counteract adverse effects of allostatic load on target organ systems [145,146].

Treatment for childhood cancer must prioritise the eradication of the cancer itself; however, secondary considerations must be given to reducing burden of illness during survivorship and improving quality of life and function. To this end, interventions that support stress buffering behaviours may have a beneficial impact on childhood cancer patient and survivors. While clinicians may already promote these types of support for their patients, understanding that these benefits not only improve experiences and subjective quality of life but are likely to have clinical implications for long-term health and wellbeing is critical to ensuring that they are included in holistic treatment of children’s cancer.

## 8. Clinical Implications and Future Research

Stress regulation is integral to how we navigate dynamic environments in everyday life. Stress, homeostasis and allostasis are concepts that have been developed for decades, with recent attempts to quantify these states being rooted in the study of thermodynamics [147]. The thermodynamic entropy-based stress model proposes that adverse health states are caused by positive stress entropic load, while negative stress entropic load leads to a protective health state, leading to the idea that energy balance may be a crucial intervention for chronic disease [147]. The chronic or inappropriate activation of stress regulatory signaling or target organ activation contributes to wear and tear on critical organ systems and can contribute to adverse health and functional outcomes [3,8]. Even though neurological changes caused by biomediator feedback occur during excessive or prolonged stress signaling, disruption to the systems regulation can occur at any step in the pathway due to self-regulatory nature. There is evidence that childhood cancer patients may have disruption to several steps in the stress-regulatory pathway including cognitive-affective function, neurological disruption to stress regulatory brain regions, altered adrenal and endocrine function, and disrupted tissue integrity, as well as lower engagement in positive coping behaviours such as physical activity and pro-social habits. Childhood cancer patients experience an array of adverse late effects of their cancer that may be brought on by or exacerbated by dysfunctional stress regulation and adversely affect their physical and mental health. Stress regulation may be a valuable lens through which to examine these chronic morbidities in childhood cancer populations. Further research is needed to better understand acute stress reactivity and stress signaling, as well as the connections between different pathways (i.e., cognitive-affective function and stress outcomes). Larger cohort studies may be necessary to accommodate interindividual variability in stress impacts on individuals and the nature of dysfunction. Even though it is not possible to differentiate between direct and indirect effects of cancer treatment during childhood, a better understanding of how neurological, physiological and psychological disruptions during the experience of childhood cancer interact to produce late effects is important. As treatments continue to improve survival rates in this pediatric clinical population, an emphasis on understanding how to improve health and wellbeing during survivorship has emerged. Clinical recognition of stress as a model during treatment, understanding clinical implications of programs supporting positive coping behaviours—psychological, social, physical activity, may thus be timely.

## Figures and Tables

**Figure 1 cancers-13-04684-f001:**
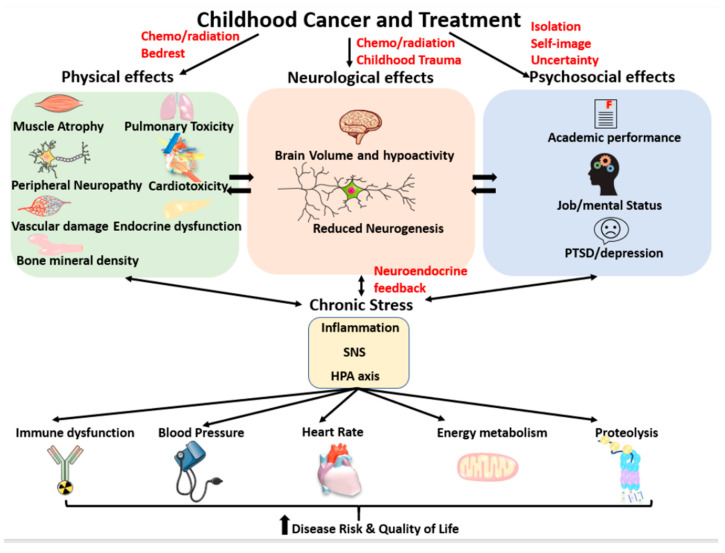
Illustration representation of the interrelationships between childhood cancer and its treatment and the physical, neurological and psychosocial effects that lead to chronic stress. A bidirectional relationship exists between physical, neurological and psychosocial effects, as well as between chronic stress and dysfunction in these systems. The major health related effects of chronic stress leading to disease risk and lower quality of life.

**Figure 2 cancers-13-04684-f002:**
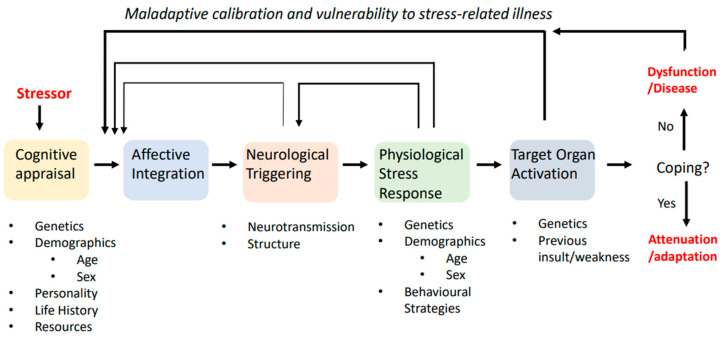
A systems approach to stress responses. Some stimulus—internal or external, perceived or real, past, present or future—is interpreted as threatening through cognitive appraisal and integrated with affective feedback. Together, the cognitive-affective appraisal determines the saliency of the threat. If sufficient, this will cause neurological triggering of stress activation pathways in the brain, resulting in increased biochemical signaling of these pathways and their subsequent physiological activation of target organs. If a stable state cannot be achieved or maintained to meet the demands of the stressor, then excessive activation and associated wear on target tissues may result and possibly lead to dysfunctional signalling with excessive feedback by mediators on earlier pathway steps. Adapted with permission from ref. [14]. Copyright 2012 Springer.

**Table 1 cancers-13-04684-t001:** Stress regulation consequences of cancer treatment on paediatric populations.

Author, Year	Purpose/Aim	Design/Methods	Participants/Sample	Findings	Limitations
**Neurocognitive outcomes and cognitive-affective appraisal**					
Brinkman (2012) [20]	To investigate the relationship between white matter and cognitive processes in adult survivors of childhood medulloblastoma.	Cross-sectional study Participants underwent neurocognitive testing and MRI (diffusion-tensor imaging)	*n* = 20 Participants were survivors of MB treated at St. Jude’s	Neurocognitive impairment was common in many domains of function. Reduced white matter integrity was associated with worse performance on tasks of executive function.	Lack of control group mall sample size Patients were treated with outdated methods for MB treatment
Brown (1996) [21]	To investigate the intellectual and academic functioning of children with leukemia treated with intrathecal chemotherapy when compared to cancer patients not treated with CNS prophylaxis.	Prospective cohort study Intellectual assessments and academic achievement tests conducted at baseline and at years 1, 2 and 3 following diagnosis.	*n* = 38 children with leukemia (receiving CNS prophylactic chemotherapy) *n* = 25 children with other cancers (not treated with CNS prophylaxis) Participants were diagnosed at the Women’s and Children’s Hospital	3 years after diagnosis, the CNS treated children performed worse on academic tests of reading, spelling and arithmetic.	Lack of healthy control group Small sample size and high attrition rate
Davidson (2000) [22]	To investigate neurological changes in cancer patients treated with high-dose MTX compared to healthy controls.	Cross-sectional study Cognitive assessment and proton magnetic resonance spectroscopy/MRI in cancer patients and healthy controls.	*n* = 11 children with ALL, non-Hodgkin lymphoma, or osteosarcoma, undergoing MTX treatment. *n* = 17 healthy controls	The choline/water ratio was lower in patients than controls. Abnormal white matter was observed in 3 cancer patients (and potentially a 4th).	Due to the small sample size, any differences in metabolites would need to be large to be detected
Kadan-Lottick (2009) [23]	To evaluate differences in neurocognitive and academic performance in children with ALL treated with DX versus prednisone.	Cross-sectional study Patients were previously randomised to corticosteroid treatment. Patients underwent a half-day neurocognitive assessment.	*n* = 51 DX-treated patients *n* = 41 prednisone-treated patients Participants were previously enrolled in Children’s Cancer Group 1991 trial.	The only significant difference between groups was on a test of reading (DX-treated scored worse)	Lack of control group
Matsuoka (2003) [26]	To assess structural differences in the amygdala of cancer survivors with/without intrusive recollections.	Cross-sectional study MRI and volumetric analysis of the amygdala was performed in both groups.	*n* = 35 breast cancer survivors with intrusive recollections *n* = 41 breast cancer survivors with no intrusive recollections Participants recruited from the outpatient clinic of the Division of Breast Surgery, National Cancer Center Hospital East	Amygdala volume was lower in participants with intrusive recollections even after controlling for age, height and depression diagnoses.	Lack of healthy control group Risk of recall bias Early life stressors were not evaluated
Krull (2013) [35]	To investigate predictors, patterns and rates of neurocognitive impairment in adult survivors of childhood ALL decades after treatment.	Cross-sectional study Participants underwent neurocognitive testing and completed a self-rating questionnaire.	*n* = 567 Participants were ALL survivors from the St. Jude lifetime cohort study.	Participants treated with chemotherapy exhibited impairment across all neurocognitive domains. Risk for executive function problems increased with survival time (cranial radiation therapy dose-dependent).	Lack of control group Did not adjust for SES Dose variability now may not reflect that of decades ago
Reddick (2014) [67]	To prospectively validate reduced white matter volume, its influential factors and neurocognitive impairments in childhood cancer survivors	Cross-sectional study MRI, volumetrics and neurocognitive testing on participants.	*n* = 383 childhood cancer survivors (199 ALL, 184 brain tumor) *n* = 67 healthy siblings	Brain tumor survivors had lower white matter volume than ALL survivors, who were lower than controls, this was associated with treatment parameters. Childhood cancer survivors performed worse than controls neurocognitive tests.	Limited area in which white matter volume was quantified, was used to assess total tissue volume for a specific anatomical region Cross-sectional design limits temporal data
Wolfe (2013) [68]	To assess the relationship between cardiorespiratory fitness and executive functioning in radiation-treated pediatric cancer survivors.	Cross-sectional study Participants underwent fMRI imaging while completing an n-back test and cardiorespiratory fitness testing on a cycle ergometer.	*n* = 9 childhood posterior fossa tumor survivors Participants were recruited from the neuro-oncology clinic at the Children’s Hospital of Alabama	Higher cardiorespiratory fitness was associated with increased working memory and efficiency of neural functioning in pediatric cancer survivors.	Small sample size and lack of controls: limited power and generalisability Correlational data limits the inference of causality Cerebellar activation was not assessed due to heterogeneity of lesions in participants
Stefancin (2020) [69]	To explore the association between chemotherapy and working memory function in childhood cancer survivors and healthy controls.	Cross-sectional study fMRI was acquired while participants performed a visual n-back test.	*n* = 15 pediatric cancer survivors, patients at the Stony Brook Children’s Hospital *n* = 15 healthy controls	Working memory impairment was present in pediatric cancer survivors when compared to controls. In survivors, correct responses generated a decreased BOLD response in the posterior cingulate, incorrect responses generated a greater BOLD response in the angular gyrus, and no response generate a greater BOLD response in the superior parietal lobule.	Cross-sectional design Increased BOLD signal could either indicate increased activation, or decreased neuronal efficiency Heterogeneity of sample in terms of cancer type and treatment
van der Plas (2021) [71]	To investigate the prevalence of neurocognitive impairments in survivors of childhood ALL and if age at diagnosis, chemotherapy, and chronic conditions correlate with risk of impairment.	Cross-sectional study Participants completed the Childhood Cancer Survivor Study Neurocognitive Questionnaire. Neurocognitive impairment associations with treatment exposures and chronic conditions were examined.	*n* = 1207 survivors of ALL *n* = 2273 siblings Participants were enrolled in the Childhood Cancer Survivor Study	ALL survivors reported increased impairments in memory and task efficiency when compared to healthy controls. In male survivors, impairments in memory were associated with increased dosage of MTX and DX exposure, while impairments in task efficiency were associated with neurologic and pulmonary conditions. In female survivors, endocrine conditions were associated with higher risk of impairments in memory and task efficiency.	Self-report of cognitive impairment Risk of participation bias Many other factors that were not studied might be associated with neurocognitive impairment
Williams (2020) [72]	To examine if childhood cancer survivors with injuries to the brain are at a higher risk for chronic health conditions and if this is associated with neurocognitive impairment later in life.	Cross-sectional study All participants completed neurocognitive testing and a clinical examination.	*n* = 2859 adult survivors of childhood cancer: 1598 had CNS therapy Participants were treated at St Jude’s and enrolled in the St Jude Lifetime Cohort Study	Participants that were CNS-treated performed worse than those that were not CNS-treated on neurocognitive testing and had more global neurocognitive impairments. There was a dose-dependent association between severity/burden of treatment and global impairment in CNS-treated participants. Chronic health conditions such as cardiovascular and pulmonary conditions were associated with impairments in memory, processing speed and attention in CNS-treated participants with neurological conditions.	Lack of healthy control group Lack of neurocognitive data immediately following treatment in survivors
Lesnik (1998) [73]	To assess frontal-cerebellar morphological characteristics and function in survivors of childhood ALL that were treated with intrathecal MTX, while using an effect size model to increase validity in a small sample.	Cross-sectional study Neuropsychological testing and MRI of cerebellar lobuli (I-V and VI-VII) and prefrontal cortices was assessed in participants.	*n* = 10 childhood survivors of ALL *n* = 10 age, sex, and socioeconomic status matched healthy controls	There were deficits in neuropsychological testing and morphometric and functional characteristics of cerebellar lobuli and prefrontal cortices in MTX-treated childhood ALL survivors. Evidence supported the involvement of the hypothesised subsystem; the cerebellar-frontal system.	Correlational data limits the inference of causality Small sample size
Peng (2021) [77]	To investigate behavioural and neurocognitive functioning in survivors of childhood ALL and evaluate the associated clinical and socio-environmental factors.	Prospective, cross-sectional Participants completed neurocognitive testing and self-reported emotional, behavioural and socio-environmental variables via questionnaires and checklists. Chronic health conditions and clinical variables were pulled from patient charts.	152 survivors of childhood ALL: 32 received cranial radiation therapy 120 received chemotherapy Participants were patients at the Long-term Follow-up Clinic of the Prince of Wales Hospital	Lower levels of self-reported physical activity were correlated with inattention and sluggish cognitive tempo. A minority of survivors had impairments in motor-processing, and attention, and developed treatment related chronic conditions.	Small sample size of cranial radiation therapy group may have eliminated differences in neurocognition between groups Risk of sampling bias Lack of healthy control group Self-report data Incomplete patient records
Harila-Saari (1998) [91]	To evaluate changes in MRI scans of the brain of childhood-treated ALL survivors and correlate the observed abnormalities with neuropsychological impairments.	Prospective cohort study MRI immediately after the cessation of treatment and 5 years later, as well as neuropsychological testing (various tests) was conducted.	*n* = 32 Participants were childhood survivors of ALL: 15 chemotherapy-treated 17 combined chemotherapy and cranial radiation-treated Participants were patients at the Department of Pediatrics at the University of Oulu	Abnormalities in MRI were heterogeneous and infrequent among participants and did not correlate with neuropsychological function. Most participants did have neuropsychological impairments, however.	Small sample size, limited statistical power
Iuvone (2002) [92]	To investigate correlations between cognitive measures and abnormalities in MRI and computerized tomography scans of childhood ALL survivors.	Prospective cohort study Cognitive testing (various tests) and prospective MRI and computerized tomography imaging were conducted once a year for 4 years.	*n* = 21 Participants were children with ALL who received CNS prophylaxis (cranial irradiation and intrathecal MTX). Participants were patients at the Division of Pediatric Oncology, Catholic University	Abnormalities in white matter were associated with poor performance on a task of visual motor integration in approximately half of the participants. Intracerebral calcifications were correlated with MTX doses, and impaired cognitive testing. Females were more vulnerable to the treatment effects.	Small sample size
Monje (2013) [94]	To explore the correlates of dysfunctional episodic memory in CNS prophylaxis-treated survivors of childhood ALL.	Cross-sectional study Participants episodically encoded visual scenes and underwent fMRI while completing a memory paradigm.	*n* = 10 CNS prophylaxis and chemotherapy-treated adult survivors of childhood ALL, patients at the Dana Farber Cancer Institute *n* = 10 age matched controls	Survivors of childhood ALL demonstrated altered BOLD signal and atrophy in the hippocampus, and poor recognition memory when compared to controls. Unsuccessful encoding in ALL survivors showed increased hippocampal BOLD signal. Differences in memory among ALL survivors was related to the magnitude of BOLD response in areas responsible for successful encoding.	Small sample size
Spitzhüttl (2021) [96]	To investigate gray and white matter volume in childhood cancer survivors and the relationship to cognitive processes.	Cross-sectional study MRI T1 weighted images were acquired for voxel-based morphometry and cognitive and fine motor coordination assessments were completed by all participants.	*n* = 43 childhood cancer survivors (non-CNS cancer), treated at the University Children’s Hospital Bern or the University Children’s Hospital Zurich *n* = 43 healthy controls	Amygdala and dorsal striatum white and gray matter volume were lower in cancer survivors. Fine motor coordination of the right hand and executive function was poorer in survivors, although still within the normal range.	Cross-sectional design, risk of cohort effects Correlations were performed for each ROI and variable separately
Phillips (2020) [97]	To examine the association between glucocorticoid and MTX treatment and disruptions to the cerebello-thalamo-cortical network and antioxidant system in the brain of survivors of childhood ALL.	Cross-sectional study Brain volumes, neurocognitive testing, functional and effective connectivity, and the association between MTX and DX treatment and neurocognitive outcomes were assessed in childhood ALL survivors and healthy controls.	*n* = 176 childhood ALL survivors, recruited from the St Jude Children’s Research Hospital Total Therapy Study XV *n* = 82 age and SES matched healthy controls from the community	Survivors had decreased cerebellar volumes compared to controls, which was associated with DX exposure. In females, effective connectivity disruption was associated with poorer executive function.	Controls did not complete neurocognitive testing Biomarkers were not available to assess oxidative injury pre-treatment associated with genetics or disease Risk of confounding effects of cytarabine on brain volume Not a representative population of all ALL chemotherapy treated patients
Carey (2008) [98]	To evaluate differences in white and gray matter between ALL survivors and healthy controls.	Cross-sectional study T1 weighted MRI images, and subsequent voxel based morphometry, and neuropsychological evaluations were acquired from participants.	*n* = 9 long term ALL survivors treated with chemotherapy, patients at the University of Arizona Pediatric Hematology/Oncology Late Effects Clinic *n* = 14 healthy controls	ALL survivors had reduced white matter in the right frontal lobes and performed worse on tests of math, attention, visual-construction skills and mental flexibility when compared to controls. Neurocognitive impairments were associated with regional decreases in white matter volume.	Small sample size Risk of confounding factors
Reddick (2006) [100]	To assess differences in neurocognitive functioning and its relationship with white matter volume in survivors of childhood ALL when compared to healthy controls.	Cross-sectional study MRI imaging, and subsequent voxel based morphometry, as well as neurocognitive tests of academics, intelligence and attention, were performed on ALL survivors and controls.	*n* = 112 ALL survivors *n* = 33 healthy siblings	Survivors of ALL performed significantly worse on tests of attention and had decreased white matter volume when compared to controls. Decreased white-matter volume was associated with impaired academics, intelligence and attention.	Limited area in which white matter volume was quantified was used to assess total tissue volume for a specific anatomical region Cross-sectional design limits temporal data
Stefanski (2020) [101]	To examine neurocognitive and psychosocial outcomes in adult survivors of childhood leukemia that were treated with bone marrow transplantation or intensive chemotherapy.	Cross-sectional study Participants completed questionnaires on emotional distress, neurocognitive problems, social attainment and health-related quality of life.	*n* = 482 adult survivors of AML: 183 bone marrow transplantation-treated 299 intensive chemotherapy-treated *n* = 3190 siblings Participants were enrolled in the Childhood Cancer Survivor Study	Survivors had greater impairments in health-related quality of life, emotional distress and neurocognitive functioning than siblings. Survivors had greater risk for unemployment, lower education and income, and not having a partner.	Lack of differences in treatment groups may have been due to sample sizes and limited power Risk of participation bias Self-report Siblings may not be representative of the general population
**Biological/cellular aging and inflammatory outcomes**					
Kennedy (2004) [35]	To assess the effects of ALL treatment in children on antioxidant status and the association between antioxidant stress, oxidative stress and complications.	Prospective cohort study At baseline (diagnosis), 3 months and 6 months, antioxidant plasma concentrations, total antioxidant capacity and DNA oxidised base 8-oxodeoxyguanosine were assessed.	*n* = 103 newly diagnosed children and adolescents being treated for ALL	Plasma vitamin A, antioxidants, total antioxidant capacity and DNA oxidised base 8-oxodeoxyguanosinconcentrations changed over 6 months. Beneficial associations were found between higher concentrations and various treatment dose parameters. Adverse relationships were also found.	Criteria for deficiency states may be limited, i.e., children with leukemia might have higher requirements
Mazur (2004) [32]	To evaluate serum levels of cytokines in children after treatment for ALL was finished.	Cross-sectional study Serum concentrations of cytokines measured using an enzyme linked immunosorbent assay.	*n* = 30 healthy controls 5 groups of 30 ALL patients: 1, 3-, 6-, 9- and 12-months post-treatment (*n* = 150 total), treated at the Department of Pediatric Hematology and Chemotherapy, Zabrze	There were significant differences in interleukin-8, tumor necrosis factor-alpha and interleukin-2 serum concentrations between ALL patients and healthy controls.	Cross-sectional design limits the inference of causality
Papageorgiou (2005) [102]	To compare TAC and corrected TAC between cancer free children and children with malignancy at the time of diagnosis and during chemotherapy.	Cross-sectional study All children were under a free diet during the study. TAC and corrected TAC levels were evaluated from blood samples.	*n* = 20 children with malignancy, recruited from the University Hospital of Heraklion *n* = 80 control participants	TAC and corrected TAC decrease progressively during cycles of chemotherapy in children with malignancy.	Small sample size Different treatment regimes used among patients Variability in patient diet and other potential confounding variables
Hasan (2020) [103]	To assess differences in serum TOS, TAC and the OSI of ALL and AML patients compared to healthy controls.	Cross-sectional study Erel’s methods were utilised to assess TOS and TAC, and OSI was calculated in leukemia patients and controls.	*n* = 60 leukemia patients, patients at the Hereditary Hematology Center *n* = 70 age and gender matched healthy controls	TOS and OSI were significantly higher in leukemia patients when compared to controls, and antioxidant levels were significantly lower. Oxidative stress was present in both ALL and AML.	
Vatanen (2017) [104]	To analyse the prevalence of frailty and physical health limitations among long-term survivors of high-risk neuroblastoma and to investigate whether frail health is associated with markers of inflammation and telomere length.	Cross-sectional study Frailty is defined as 3 or more of the following: low lean muscle mass, low energy expenditure, slowness, weakness, exhaustion. Cardiovascular function and telomere length analysis were also performed.	*n* = 19 cancer survivors *n* = 20 healthy controls	Prevalence of frailty was significantly higher in survivors versus controls (47% vs. 0%). 68% or survivors reported limitations in vigorous activity versus 0% of controls. Survivors had significantly shorter telomeres and significantly higher CRP levels.	Small sample size Definition of frailty not wholly comprehensive
Song (2020) [105]	To analyse and compare leukocyte telomere length and age-related attrition between childhood cancer survivors and non-cancer controls. Leukocyte telomere lengths were also analysed for association with treatment exposures, chronic health conditions and health behaviours among survivors.	Retrospective cohort study with prospective clinical follow-up. Leukocyte telomere length was measured using whole genome sequencing. Common non-neoplastic health conditions and subsequent malignant neoplasms were clinically assessed.	*n* = 2427 childhood cancer survivors, recruited from St. Jude’s Children’s Hospital *n* = 293 non-cancer controls	Leukocyte telomere length was significantly shorter in childhood cancer survivors compared to non-cancer controls. Shorter leukocyte telomere length was correlated with specific treatments including chest and abdominal irradiation, glucocorticoid and vincristine chemotherapies.	Risk of confounding bias Correlational data
Qin (2021) [106]	To evaluate EAA and its association to chronic health conditions, health behaviour and treatment exposures in survivors of childhood cancer.	Cross-sectional study Methylation data was generated from cancer survivors and controls. EAA was calculated as residuals from a linear regression of epigenetic age and chronological age. EAA adjusted least square mean was compared across health behaviours and treatment exposures. The associations between EAA and 20 different chronic health conditions was assessed.	*n* = 2139 childhood cancer survivors *n* = 282 frequency matched controls Participants were enrolled in the St Jude Lifetime Cohort Study	EAA was greater in childhood cancer survivors than in controls. Among survivors, higher EAA was observed in patients that were previously various cancer treatments. Associations between several chronic health conditions hypertension, myocardial infarction, obstructive pulmonary deficit, peripheral motor and sensory neuropathy, and pulmonary diffusion deficits and EAA were observed.	Limited power due to few participants having specific chronic health conditions and non-matching controls Only treatments within 5 years of diagnosis were considered Results cannot be generalised, as all participants were of European descent Temporal associations of health behaviours and EAA were unavailable
Ravera (2021) [107]	To investigate the molecular and metabolic markers of early aging in survivors of childhood cancer.	Cross-sectional study Mononuclear cells were isolated from the blood of childhood cancer survivors and healthy controls, and assessed using biochemical, proteomic and molecular biology analyses.	*n* = 196 childhood cancer survivors *n* = 154 healthy controls	Survivors had an increased biological age by decades compared to their chronological age. Survivors had inefficient oxidative phosphorylation which was associated with decreased energy and the switch to lactate fermentation, increased lipid peroxidation and decreased expression of genes/proteins involved in metabolism and mitochondrial biogenesis.	Cross-sectional study design
Hayek (2020) [108]	To investigate the prevalence of frailty in survivors of childhood cancer, and its association with cancer treatment and other factors.	Retrospective cohort study Participants completed a baseline and follow-up questionnaires. A generalised linear model evaluated associations between frailty, treatment and other variables.	*n* = 10,899 childhood cancer survivors *n* = 2097 siblings Participants were enrolled in the Childhood Cancer Survivor Study	Survivors had increased frailty compared to siblings. Radiation treatment and lung surgery were associated with increased risk of frailty for survivors.	Lack of participation from all eligible subjects may inflate/deflate prevalence estimates Risk of recall bias Risk of survival bias

Abbreviations: MRI, magnetic resonance imaging; MB, medulloblastoma; CNS, central nervous system; ALL, acute lymphoblastic leukemia; MTX, methotrexate; DX, dexamethasone; AML, acute myelogenous leukemia; fMRI, functional magnetic resonance imaging; BOLD, blood-oxygen-level-dependent; DNA, Deoxyribonucleic Acid; TAC, total antioxidant capacity; TOS, total oxidant status; OSI, oxidative stress index; CRP, c-reactive protein; EAA, epigenetic age acceleration. Table 1 inclusion criteria: A scoping review of the literature was completed examining: (1). Primary research articles examining structural and functional neurological impairments in pediatric cancer patients; (2). Primary research articles examining inflammatory consequences and biological or cellular aging in pediatric cancer patients; (3). Pediatric cancer patients or survivors of childhood cancer (pediatric and adult).

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
