# Peer review of "The Psychoneuroimmunology of Stress Regulation in Pediatric Cancer Patients"

_cancers, 2021, doi:10.3390/cancers13184684_

Round 1

Reviewer 1 Report

The psychoneuroimmunology of stress regulation in pediatric cancer patients  

In their manuscript, White and coworkers discuss the possibility that childhood illness (and in particular cancer) constitutes an episode of early life stress comparable to already considered Adverse Childhood Events. While the originality of this idea certainly is much appreciated and should be further explored, I feel that manuscript should be improved in many ways detailed below.

Major comments

1) Title and throughout manuscript: It is neither explained what psychoneuroimmunology is nor argued why this domain would be more fruitful for the question at hand than, for example, psychoneuroendocrinology.

2) It should be discussed that there are at different ways to consider allostasis. Furthermore, the authors should argue and make explicit why they favor a vision of allostasis as ‘physiological stress regulation”. 

3) The usefulness of animal models for childhood cancer should be discussed.

4) The potential role of the parents’ responses to the stress of having a child with cancer should be discussed with respect to the potential consequences for children.

5) The authors are invited to discuss the distinction between so-called psychological and physiological (hypoglycemia, infection, tissue injury and tumor growth, for example) stressors and to revise figure 2, which seems to mostly concern psychological stressors, accordingly.

6) Some of the reasoning (on page 5, lines 172-176; page 6, lines 220-240) is rather circumstantial. Please, render more precise or otherwise drop.

7) The inclusion criteria for the articles included in Table 1 are unclear and most of them do not seem to have a primary interest in or relevance for stress. Please, provide criteria and redo this table.

8) The authors give the impression that corticolimbic structures is a clearly circumscribed entity with straightforward structure-function relationships. Nothing could be further of the truth, according to me. Please, include a discussion on what the corticolimbic structures comprise and the many different functions (other than in response to stress) that these structures have been associated with. 

Minor comments

9) Page 4, lines 104-106: “If a stable state cannot be achieved or maintained to meet the demands of the stressor, then disease/dysfunction will result.” This is too general of a statement as some diseases are precisely due to a too stable state and an inherent less of flexibility. Please, rephrase.

10) Page 21, lines 263-264: “The biomediators of inflammation, cytokines, can be produced by catecholamine demargination and activation of immune cells [86].” Please explain what “demargination” means or use a different term.

11) Page 21, lines 279-281: “Elevated inflammation is also a common finding that is a purported mechanism of long-term adverse health and functional outcomes in chronic stress populations [31].” This has indeed been proposed, but it would be good to indicate that the inflammatory response to infection and that to psychological stress are very different in degree and probably in nature.

Author Response

We thank the reviewer for their feedback. Please see our responses in the attachment.

Reviewer 2 Report

24 August 2021

Review on the manuscript titled “The psychoneuroimmunology of stress regulation in pediatric cancer patients” by White GE et al., submitted to Cancers

Manuscript ID: cancers-1342258

Dear Authors,

Children with cancer show similar changes in stress regulation and long-term effects to those who experience early-life stress. The changes include cognitive, affective, neurological, endocrine, and renal functions and coping behaviors. However, little is studied about stress reactivity in childhood illness. The authors review psychoneuroimmunology and differences in stress regulation among childhood cancer patients, proposing that understanding stress regulation in childhood cancer is important and trying to reveal a clinically relevant stress pathway in search of potential interventional sites.

Please reconsider the following:

  1. A graphical abstract summarizing the manuscript is highly recommended.
  2. Page 1, Keywords: Please list up to ten keywords.
  3. Pages 1,2, Introduction, Paragraph 1: Stress is a state of the body`s adaptational response to stressors and the stress mediators may damage the host. Please clearly describe stress, stressors, and the cause of consequence.
  4. Pages 3,4, The Section 2: Allostasis refers to the process of maintaining stability through change. Please make clear the difference between homeostasis, Selye’s Stress, and allostasis in the beginning of the section.
  5. Pages 3-24, The Sections 2-6: Short descriptions unique to the developmental stage of children in each section surely increase the value of this manuscript.
  6. Page 4, Figure 2: The cognitive appraisal may include anticipation.
  7. Page 4, The Section 2, Paragraph 4: Please describe “allostatic load” clearly.
  8. Page 7-21, Table 1: Please divide the table according to the subheadings and present them in each subsection.
  9. Page 22, Figure 3: Please present the figure in color.
  10. Pages 22,23. The Section 6: The long-term immune tolerance and low-grade inflammation is considered to be one of allostatic loads. Suggested refences: https://www.mdpi.com/2227-9059/9/7/734.
  11. Pages 24,25, The Section 8: The historical aspects of stress and attempts to quantify stress are discussed recently. Suggested reference: DOI: 10.17219/acem/139572.
  12. Pages 25-29, References: Please cite more articles, preferable more than 150 for review articles.

The manuscript contains two figures, one table, and 122 references. The manuscript carries important value presenting consequences of stressful events in psychoneuroimmunology of children with cancer and possible interventional targets. I recommend this manuscript for publication after major revision.

Author Response

We thank the reviewer for their report and feedback. Please see our responses in the attachment.

Round 2

Reviewer 1 Report

The authors have dealt adequately with almost all of the points of criticism raided.

Author Response

We would like to thank you for your comments and suggestions.

Reviewer 2 Report

24 August 2021

Review on the manuscript titled “The psychoneuroimmunology of stress regulation in pediatric cancer patients” by White GE et al., submitted to Cancers

Manuscript ID: cancers-1342258

Dear Authors,

The manuscript contains tree figures, one table, and 148 references. The manuscript is revised accordingly and thus, the quality is substantially improved.

A graphical abstract is highly recommended as it certainly attracts more attention of readers and increases the chance of citation. The graphical abstract is for online readers and it will not be in PDF version. So, it deserves to present it, even though it becomes similar to one of figures in the manuscript.  Please color Figure 3, as in Figures 1 and 2. The Journal will not charge.

The manuscript carries important value presenting consequences of stressful events in psychoneuroimmunology of children with cancer and possible interventional targets. I recommend this manuscript for publication after minor revision.

Best regards,

Author Response

Thank you for your suggestions. We have added a graphical abstract to the manuscript. We have also revised Figure 3 to be presented in color.